# Assessing the Risk of Occurrence of Bluetongue in Senegal

**DOI:** 10.3390/microorganisms8111766

**Published:** 2020-11-11

**Authors:** Marie Cicille Ba Gahn, Fallou Niakh, Mamadou Ciss, Ismaila Seck, Modou Moustapha Lo, Assane Gueye Fall, Biram Biteye, Moussa Fall, Mbengué Ndiaye, Aminata Ba, Momar Talla Seck, Baba Sall, Mbargou Lo, Coumba Faye, Cécile Squarzoni-Diaw, Alioune Ka, Yves Amevoin, Andrea Apolloni

**Affiliations:** 1Institut Sénégalais de Recherches Agricoles, Laboratoire National de l’Elevage et de Recherches Vétérinaires (ISRA-LNERV), Dakar-Hann BP 2057, Senegal; mariececille.gahn@gmail.com (M.C.B.G.); fallou.niakh@ensae.fr (F.N.); ciss.mamadou@gmail.com (M.C.); moustaphlo@yahoo.fr (M.M.L.); agueyefall@yahoo.fr (A.G.F.); biteye88@yahoo.fr (B.B.); moussafall08@yahoo.fr (M.F.); mbenguallb@gmail.com (M.N.); baaminata10@gmail.com (A.B.); mtseck@hotmail.fr (M.T.S.); aliouneka95@outlook.fr (A.K.); amevoin@gmail.com (Y.A.); 2ASTRE, Univ Montpellier, CIRAD, INRAE, F-34398 Montpellier, France; cecile.squarzonidiaw@cirad.fr; 3Centre de Coopération Internationale en Recherche Agronomique pour le Développement (CIRAD), UMR ASTRE, F-34398 Montpellier, France; 4École Nationale de la Statistique et de l’Administration Économique, 91764 Palaiseau CEDEX, France; 5FAO, ECTAD Regional Office for Africa, 2 Gamel Abdul Nasser Road, P.O. Box GP 1628, Accra, Ghana; ismaila.seck@fao.org; 6Direction des Services Vétérinaires, Dakar 45677, Senegal; babasall@hotmail.com (B.S.); drmbargoulo@gmail.com (M.L.); coumba.diouf@fao.org (C.F.); 7CIRAD, UMR ASTRE, F-97491 Ste-Clotilde, La Reunion, France

**Keywords:** vector abundance model, compartmental model, basic reproduction number, risk map

## Abstract

Bluetongue is a non-contagious viral disease affecting small ruminants and cattle that can cause severe economic losses in the livestock sector. The virus is transmitted by certain species of the genus *Culicoides* and consequently, understanding their distribution is essential to enable the identification of high-risk transmission areas. In this work we use bioclimatic and environmental variables to predict vector abundance, and estimate spatial variations in the basic reproductive ratio $\;R_{0}$. The resulting estimates were combined with livestock mobility and serological data to assess the risk of Bluetongue outbreaks in Senegal. The results show an increasing abundance of *C. imicola*, *C. oxystoma*, *C. enderleini*, and *C. miombo* from north to south. $R_{0}$ < 1 for most areas of Senegal, whilst southern (Casamance) and southeastern (Kedougou and part of Tambacounda) agro-pastoral areas have the highest risk of outbreak ($R_{0}$ = 2.7 and 2.9, respectively). The next higher risk areas are in the Senegal River Valley ($R_{0}$ = 1.07), and the Atlantic coast zones. Seroprevalence rates, shown by cELISA, weren’t positively correlated with outbreak probability. Future works should include follow-up studies of competent vector abundancies and serological surveys based on the results of the risk analysis conducted here to optimize the national epidemiological surveillance system.

## 1. Introduction

Bluetongue (BT) is a non-contagious viral arthropod-borne disease of domestic and wild ruminants. The disease is caused by the Bluetongue virus (BTV), a member of the genus Orbivirus. Bluetongue is listed as an important transboundary animal disease by the World Organization for Animal Health (OIE) [1], and its occurrence in animals must be notified to the OIE. BTV is biologically transmitted by females of several species of *Culicoides* (Diptera: Ceratopogonidae) biting midges. The clinical signs of Bluetongue infection can include fever, edema of the lips, tongue, and head, conjunctivitis, nasal discharge, and pain at mucocutaneous junctions. A series of studies have also provided evidence of BTV-8 transplacental transmission in experimental infections in cattle, sheep, and goats [2,3,4], and in-field observations of aborted fetuses [5]. There is also evidence that infection in the early stages of fetal development may result in abortion [6]. 

Bluetongue is enzootic in many tropical, subtropical, and some temperate regions, including much of the Americas, Africa, South Asia, and northern Australia, coincident with the geographic distribution and seasonal activity of competent *Culicoides* vector insects [7,8,9]. Bluetongue virus can be introduced in distant areas through the dispersion of infected *Culicoides* by moderate winds (10–40 knm/h) and/or by movements of infected animals. The first official report of BT emanated from the Cape Province of South Africa in the late eighteenth century, following the importation of susceptible Merino sheep from Europe [10], while the first confirmed outbreak of Bluetongue disease outside Africa occurred in Cyprus [11]; subsequently, several outbreaks of the disease were reported in different parts of the world including Israel, Pakistan, and India [12]. In the last five months of 2006, BTV-8 outbreaks occurred almost simultaneously in Belgium, France, Germany, and the Netherlands [13,14]. At the end of 2007, 14,264 premises were listed in France, and more than 30,000 cases were reported in 2008 [15]. 

The existence of BTV is little known in African countries. In most of the cases, local breeds of sheep and goats are resistant to the disease, and because of this, the disease is not always clinically diagnosed [16]. In these conditions, its presence can only be proven by systematic studies: Serological surveys or BTV detections/isolations from Culicoides vectors and animals. The origin of Bluetongue in Senegal remains the subject of debate. In 1925, Curasson described outbreaks in Merino sheep imported from South Africa as responsible for the introduction of the disease in Senegal. However, he later reported that the disease had been circulating in symptom-free form throughout the continent years before 1925 [17].

*C. imicola* and *C. oxystoma* species have been identified as the main BTV vectors in several entomological studies conducted in South Africa, North Africa, southern Europe, and India [18,19]. The roles of *C. kingi*, *C. enderleini*, and *C. miombo* as potential BTV vectors in Senegal due to their abundances, distribution, or feeding behaviors have been reported by several authors [19,20,21]. In the case of *enderleini,* some BTV seropositive specimens were collected in Sudan. In Senegal, following a serious outbreak of African horse sickness (AHS) in 2007 [22,23], several studies were conducted on the diversity and ecology of *Culicoides* vectors [18,19]. It has been shown that in sub-Saharan Africa, the same vectors that transmit BT are also responsible for the transmission of AHS. For this reason, understanding the spatial distribution of *Culicoides* species could help control the spread of both AHS and BT. Different statistical models have been developed for the *Culicoides* species involved in the transmission of BTV and AHSV [24,25]. All these models account for meteorological and environmental variables, such as rainfall, temperature, humidity, NDVI, or flooding, as key drivers.

The specific objectives of this work were to (i) model the spatial abundance of Culicoides of veterinary interest, (ii) estimate the epidemic potential of BT in Senegal, and (iii) assess the probability of an epidemic of BT being triggered in the area. Following previous studies in the area [20,21], we predicted vector abundance using bioclimatic and environmental variables to train random forest models. We used the basic reproduction number (R_0_) as a measure of BT epidemic potential and mapped its values. The transmission dynamics are described using a compartmental model in a multi-species context, and we used the Next-Generation Matrix [26] approach to determine R_0_. Finally, we combined previous results with mobility and serological data to estimate the risk of occurrence of the disease in the different parts of Senegal due to the introduction of animals from infected zones [27,28]. 

## 2. Materials and Methods

### 2.1. Study Area

For this work, we used data collected throughout Senegal. The resulting maps provide information for all the areas in Senegal. Senegal is located in West Africa between latitudes 12° and 17° N and longitudes 11° and 18° W. It is bounded on the west by the Atlantic Ocean, on the north by the Senegal River that constitutes a natural frontier with Mauritania, on the east by Mali, and on the south by the two Guineas (Guinea and Guinea Bissau) and completely surrounds the Gambia. Senegal is divided into 14 administrative regions (Figure 1) and 45 departments. Six agro-ecological zones are defined based on their climatic and environmental characteristics: Niayes, River Senegal Valley, Ferlo, Eastern Senegal, Casamance, and Groundnut Basin. Two pastoral zones are defined based on animal production: The sylvo-pastoral in the north and the agro-pastoral in the (Figure 1).

### 2.2. Data Collection

#### 2.2.1. Entomological Data

As part of a nationwide entomological surveillance program in Senegal in 2012, 108 livestock premises were initially selected (as trapping sites): Three sites per department and three departments per region in 12 out of a total of 14 regions in Senegal. The sites were selected to better understand the complex relationship between host-pathogen-vectors and the role of environmental and climatic factors in the emergence and/or re-emergence of vector-borne diseases, such as BT and AHS. Among the 108 originally selected sites, 98 sites were sampled while 10 sites were not due to logistic problems. In two of the 98 sampled sites, the trapping results could not be exploited, and were thus, discarded (one conservation problem and one battery failure). In this study, we used only data from 96 sites that were visited at the end of the rainy season in 2012 (between September and October); their location is shown in the Appendix A.

Culicoides species were collected using Onderstepoort black-light suction traps (Onderstepoort Veterinary Institute South Africa) as detailed in Reference [20]. Culicoides species were identified, as explained in Reference [29]. A maximum abundance of two nights trapping was considered as the best estimate of the population present, as abundance can decrease rapidly in sub-optimal trapping conditions [21]. The geographical coordinates of each site were recorded with a Garmin© hand-held global positioning system receiver (accurate to within 10 m) and projected in UTM Zone 28 N. The species distribution is shown in Figure 1; each box shows the distribution of the species, the size of each box is proportional to the number of Culioides collected.

A total of 1,373,929 specimens of the genus *Culicoides* belonging to 32 different species were collected in 96 sites. Almost half of these specimens belonged to the four species of veterinarian interest. Only a few culicoides were collected in the northern area regions of Saint Louis, Louga, and Matam, the majority being collected in the regions of Dakar, Kolda, and Fatick. *C. oxystoma*, *C. enderleini*, *C. imicola*, and *C. miombo*, were the four most abundant species among species of veterinary interest. In the 96 sites sampled, the percentages of individuals caught compared to the total catches were 26.9% for *C. oxystoma*, 23.89% for *C. enderleini*, 14.38% for *C. imicola*, and 12.87% for *C. miombo*, respectively [20]. However, their distribution varied in space—*C. oxystoma* was the most abundant species in the Dakar and Fatick regions, and *C. enderleini*, and *C. miombo* was the most abundant in the southern regions. More information can be found in the Appendix A.

#### 2.2.2. Serological Data

An epidemiological survey was conducted in 2018 in the framework of the European Project PaleBlu in 13 of the 14 regions of Senegal (i.e., except Ziguinchor), to assess the prevalence of Bluetongue disease (BT) in the country. Two or three sampling sites were chosen in each region, giving a total of 28 sites sampled (Appendix A). During the sero-survey 1321 serum samples were collected from 827 sheep and 494 goats. The samples were stored in the freezer at −20 °C before analysis. A competitive enzyme-linked immunosorbent assay (cElisa) was used to detect antibodies against Bluetongue virus (BTV) VP7 protein. The tests were performed at the LNERV/ISRA virology laboratory using the IDvet cElisa Kit (Idvet, 310 rue Louis Pasteur, 34,790 Grabels France) [30]. Results, aggregated at the regional level, are presented in Figure 1 and in Reference [31]. The seroprevalence was found to be high in the country, with an average seroprevalence higher than 50%. This was particularly true in the northern sylvo-pastoral area, in the region of Fatick and in the agro-pastoral regions in the southeastern corner of Senegal. More information can be found in the Appendix A.

#### 2.2.3. Climatic, Environmental and Livestock Data

Climatic, environmental, and livestock variables characterizing favorable habitats for Culicoides were selected based on a literature review of presence and abundance models [20,21,25]. A total of 26 variables were selected belonging to four categories: Eleven bio-climatic variables related to temperature (Bio01–Bio11); eight bio-climatic variables concerning rainfall (Bio12–Bio19); elevation (one variable), and animal density (six variables). The bioclimatic data, with a spatial resolution of 30 arc-seconds (~1 km), were downloaded from the world climate website (http://www.worldclim.org/current) and averaged over a 50-year period (1950–2000) at the same spatial resolution. Elevation data (digital elevation model) were extrapolated from the Moderate Resolution Imaging Spectroradiometer (MODIS) with a spatial resolution of 30 arc-seconds (~1 km). The complete list of all data used in the analysis can be found in Appendix A [20,21,25]. Finally, livestock data (density of cattle, small ruminants, horses, and donkeys) was obtained from the global distribution of livestock maps produced by FAO (http://www.fao.org/ag/againfo/resources/en/glw/GLW_dens.html). All climatic, environmental, and livestock data, are provided in the form of a raster with a resolution of 250 m * 250 m. For all the other non-rasterized information—first, we linked the corresponding parameter value to each department (Administrative Unit 2); then second, we associated the value with each pixel in the Administrative Unit. All raster data were the same resolution to make them easy to combine in subsequent steps.

#### 2.2.4. Small Ruminants Demographic Data

We used demographic data on our target animals (sheep and goats) from the Baobab survey database that combines data collected during a 15-year follow-up study of goat and sheep herds in two representative pastoral areas (the sylvo pastoral and the agro-pastoral ones). A total of 79,000 animals were surveyed in Louga (sylvo-pastoral area) and Kolda areas (agro-pastoral area) where data were collected every two weeks over a period of 15 years. The dataset contains information about the major demographic events (births, deaths, purchases, and sales) in the herds and has been used to estimate mortality and birth rates [32]. In our study, the same values as those used for the demographics rates were used for all the pixels in the same agro-ecological zone. Demographic rates change over the curse of the year and with the species, but for our purposes, we used the average daily rate. Figure 1 shows the distribution of the average birth and the daily rate in the two agro-ecological zones.

#### 2.2.5. Mobility Data

Mobility data for 2013 to 2015 were collected by the Senegalese veterinarian services. In Senegal, a system of sanitary passes (laissez-passer sanitaire (LPS)) exists to monitor national livestock mobility in the area. Every time animals are moved between two locations, the herder has to declare the movement to the nearest Veterinary Office at the origin of the movement that issues the LPS, which contains the following information: Origin and destination of the movement, date, composition, and size of the herd, means of transport, stopover. A copy is given to the herder; another is stored at the local office, and a third one should be sent regularly to the central office [25]. For reasons of completeness, we only used data for the year 2014. Together with these data, we also included data collected by the ISRA on livestock movement in the northern part of Senegal after the RVF epizooties of 2013. We focused only on the movements of ruminants. The data were cleaned and geo-localized and aggregated at the department level (Administrative Unit 2), thus, providing a representation of the inter-department mobility network. In Figure 2a, we provided a pictorial representation of the livestock mobility network. Each arrow corresponds to the oriented movements between two departments, while its color varies as a function of the average volume of animal transported. For our purposes, for each department, we estimated the incoming and outgoing volume of animals—that is, the number of animals entering or leaving a department, as shown in Figure 2b,c. We noticed that a large proportion of animals headed toward the department of Saint Louis and that certain departments in the Louga and Matam regions are the locations where most animal movements originated. At the same time, the regions of Saint Louis and Louga are also the destinations of most of the livestock. These fluxes are likely linked to transhumance in the areas and to the presence of the big market of Dahra (in the Louga region). We noticed that Kaolack is also the destination of large numbers of animals.

### 2.3. Estimated Vector Abundance, R_0_, and the Probability of the Occurrence of an Outbreak

#### 2.3.1. Abundance Modeling: Random Forest

We used the random forest method to map the abundances of the species of interest. This approach can be applied to abundances by making regressions for quantitative variables (Random regression forest) [25]. The samples used for the regression trees were obtained using random bootstrapping of individuals. The prediction of the output results from the average of those obtained at the level of each regression tree. For each species (*C. oxystoma, C. enderleini, C. imicola*, and *C. miombo*), we trained a random forest (RF) model to map their abundance, using the bioclimatic and environmental variables as predictors. For our purposes, we generated 3000 trees for each species separately. The results consisted of a map of the distribution of the abundance of each species. The choice of this method was based on the best accuracy obtained in a comparison of models for abundance species used by Diarra et al. [20].

#### 2.3.2. Model Host-Vector for Bluetongue Transmission

We developed a compartmental model for host species (small ruminants) and the *Culicoides* vector to estimate $R_{0}$. The model is based on a previously published work [25,33]. Transmission can occur only between host and vectors with no vertical or horizontal transmission. The model assumes BTV is transmitted by an infected *Culicoide* biting a susceptible small ruminant, and vice versa. In the case of the host, after a successful infection, susceptible animals (S) are initially exposed (E) before showing symptoms and being capable of transmitting (I) the virus to *Culicoides* feeding on their blood. After a certain amount of time (Infectious period = 1/r_h_), animals either die (D) with a certain probability p, or recover I and acquire lifelong immunity with probability 1-p. Similarly, for the vector population, after infection, susceptible vectors (S) become latent (E), and then infectious for the rest of their lives (I). On top of the transmission dynamics, we considered the demographic dynamics, with animals and vectors being born and dying naturally. Animals born from recovered mothers are protected by maternal antibodies (Imm_h_) for the first three months of their lives before becoming susceptible. The resulting compartmental model is a system of nine ordinary differential equations reported in Appendix A. A pictorial representation of the model is shown in Figure 3, where the pedex h corresponds to the host-related compartments, and the **v** one to the vector ones.

The force of infection for the host (*λ_h_*) and for the vectors (*λ_v_*) is estimated as:(1)$$\lambda_{h}=\frac{P_{vh}a_{s}I_{v}}{N_{h}}\lambda_{v}=\frac{P_{hv}a_{s}I_{h}}{N_{h}}$$
where *P_vh_*, *P_hv_* are the probabilities of transmission from vector to host and to host to vector, respectively, $a_{s}$ is the biting rate, $I_{h},I_{v}$ the number of infectious hosts, and vectors $N_{h},N_{v}$ their populations. 

We used the next generation matrix approach [26] to formulate the basic reproduction number $R_{0}$. This approach has the advantage of both rigorous biological interpretation and exclusion of irrelevant information. This technique is widely used in epidemiology for the determination of the basic reproductive number of vector-borne diseases [34,35,36].

Without considering the vertical transmission the $R_{0}$, is:(2)$$R_{0}=a_{s}\sqrt{\frac{P_{vh}P_{hv}\alpha_{h}\alpha_{v}}{\left(\alpha_{h}+m_{h}\right)\left(\alpha_{v}+m_{v}\right)\left(r_{h}+m_{h}\right)\left(m_{v}\right)}\frac{N_{v}}{N_{h}}}$$

Epidemiological parameters were extracted from a literature review. Some vector-related parameters could be temperature-dependent. Table 1 lists both temperature-dependent and temperature-independent values of the parameters used.

To produce a transmission map of Senegal, we divided the area into pixels with a resolution of 250 m × 250 m. For each pixel, we evaluated $R_{0}$ based on the above formula using the corresponding values of the demographic and epidemiological parameters. We repeated this operation, considering both temperature-independent and temperature-dependent estimates to account for temperature variations in Senegal.

#### 2.3.3. Sensitivity Analysis

We performed a one-factor-at-a-time (OAT) sensitivity analysis varying one parameter at a time by ±10% of its original value and estimating the corresponding variation in $R_{0}$. But instead of focusing on variation in $R_{0}$, we evaluated the fraction of pixels whose $R_{0}$ value becomes larger or smaller than 1, indicating an increase or decrease in the risk of an epidemic in the area. We considered pixels with $R_{0}>1\;$ as being *at risk*.

#### 2.3.4. Occurrence Probability Map

When an infected animal is introduced into a naïve area (j), the probability of disease extinction is $1/R_{0j}$, where $R_{0j}$, is the basic reproductive ratio of the area. Consequently, the probability of triggering an epidemic by $i_{j}$ incoming infected animals [28,40] is
(3)$$\Pi_{j}=1-{\left(\frac{1}{R_{0j}}\right)}^{i_{j}}$$

Starting from this relation, we built a map of the probability of an outbreak occurring after the introduction of infected animals. The total number of infected animals during an epidemic in the area of origin (*k*) is given by the prevalence of the disease in the area of origin ($z_{k}$). Each infected animal stays in the infectious state for an average time during $1/r_{h}$, during which it can travel and infect other animals. To a first approximation, we can therefore consider the number of newly-infected animals reaching the naïve area (*j*) during the duration of an outbreak was estimated as the sum of incoming animals from other departments ($n_{jk}$) multiplied by the fraction of infectious animals:(4)$$i_{j}=\sum_{k}\frac{z_{k}n_{jk\;}}{r_{h}},$$
where the sum is extended to all the departments directly connected to the department to which the pixel (*j*) belongs. We assume that animals entering a department ($n_{jk}$) can homogeneously diffuse the disease throughout the department, and hence, reach all the raster cells in the department. The quantities ($n_{jk}$) are estimated from the mobility data, prevalence ($z_{k}$) is taken from serological data, and $r_{h}$ is the recovery rate.

## 3. Results

### 3.1. Estimating the Vector Spatial Abundance

For each species (*C. oxystoma*, *C. enderleini*, *C. imicola*, and *C. miombo*), we trained a random forest (RF) model to map their abundance, using the bioclimatic and environmental variables as predictors. The results are shown in Figure 4. Abundance maps based on RF modeling showed that the predicted abundances of all four species of veterinary interest are very low along the Senegal River Valley, but very high in the south. In addition, the predicted abundances of *C. enderleini* are very high in the southern third (Casamance area and the southeastern corner of Senegal), and high for *C. imicola* and *C. miombo*, in the southern and middle third. The predicted abundances of *C. oxystoma* are high along the west coast, in the regions of Dakar (Niayes area), Fatick (Groundnut Basin), and Ziguinchor (Casamance). 

We identified the factors driving the vector abundance in Senegal using the Increase Node Impurity (IncNodePurity) or Mean Decrease Gini: This is a measure of variable importance based on the Gini impurity index used for the calculating splits in trees. The higher is the value of the mean decrease accuracy, the more important is the variable to our model. Appendix A shows the ten most important variables for each species according to RF modeling. The analysis shows that elevation (dem), temperature seasonality (Bio04), cattle density (Cattle), and the minimum temperature of the coldest month (Bio06) are the variables that most influence the abundance of *C. imicola*, *C. oxystoma*, *C. enderleini*, and *C. miombo,* respectively. In addition to the elevation variable, the mean temperature of the wettest quarter (Bio08) and precipitation of the wettest month (Bio13) also influence the abundance of *C. imicola*. In addition, livestock density and precipitation of the warmest quarter (Bio18) contribute to the abundance of *C. oxystoma*. Precipitation of the coldest quarter (Bio19), the mean temperature of the wettest quarter (Bio08), the mean temperature of the coldest quarter (Bio11), and cattle density (Cattle) are the variables that have the least influence on the abundance of *C. imicola*, *C. oxystoma*, *C. enderleini*, and *C. miombo,* respectively.

### 3.2. Estimating the Spatial Distribution of R_0_ (Transmission Maps)

Host density and vector abundance differ from one area to another, resulting in different risks of an outbreak and persistence of an epizootic in each of these areas. We evaluated the basic reproduction number ($R_{0}$) in Senegal: We divided the Senegal area in pixels of size 250 m × 250 m and for each of them, we evaluated $R_{0}$ using the expression in Equation (2), combining the values of host and vector densities and the epidemiological parameters. We considered both temperature-independent and temperature-dependent parameters. We refer to these maps as *transmission maps.*

The two transmission maps (Figure 5) show that the reproduction number is less than 1 in the extensive areas of Senegal. The two approaches produce similar results (Temperature Independent case median $R_{0}\;=\;0.86,\;95\%\;C.I.\left[0.27;8.70\right]$; temperature-dependent case median $R_{0}\;=\;0.61,\;95\%\;C.I.\left[0.16;6.27\right]$), although in the first case, the distribution is wider. In both cases, *R_0_* is high in Casamance (mean $\;R_{0}=2.74,1.74$ for the temperature-independent and dependent case) and in the southeastern corner of Senegal (Kédougou region). Indeed, in the latter, the basic reproduction number is higher than 10. Similarly, in Casamance, the value of $R_{0}$ is high (above 5 in almost all the area)—possibly linked to the high abundance of *C. enderleini*. On the other hand, in the northern part of the country (particularly in the Ferlo area), the value of $R_{0}$ is low (mean $\;R_{0}=0.66,\;0.48$ for the temperature-independent and dependent case), meaning there is little risk of an outbreak of BT. Moreover, the map built using temperature-independent values predicts higher values of $R_{0}$ in the Casamance area, the groundnut basin *(Bassin arachidier)* (mean $\;R_{0}=0.80,\;0.51$ for the temperature-independent and dependent case) and along the Senegal River Valley in the north (mean $\;R_{0}=1.08,\;0.74$ for the temperature-independent and dependent case), than the map built using temperature-dependent values. The emerging trends reflect the distribution of *Culicoides*, and any discrepancies between the two approaches could be linked to the different biting rates and latency period. Indeed, areas with high vector abundance (Figure 4) present the highest value of $R_{0}$ due to the high vector-host ratio ($\frac{N_{V}}{N_{h}}$), while areas with low vector abundance (e.g., the Ferlo area) have the lowest risk of Bluetongue transmission.

Since small ruminant demography (in particular mortality) varies over the year, we also produced $R_{0}$ distribution maps for each month of the year (see Appendix A), but no significant variations were found.

We evaluated the sensitivity of our indicator ($R_{0})$ by increasing or decreasing one of the input parameters by 10% and monitoring the resulting variations, leaving the others at their baseline values. Instead of focusing on the rate of variation of $R_{0}$ we focused on the average variation in the number of pixels whose $R_{0}$ is above 1 (*at risk*) (Figure 6). An increase in the number of pixels *at risk* following the variation of a parameter would mean that the zone where infection could occur has spatially widened, while the area *at risk* is reduced. We performed the sensitivity analysis for both the temperature-independent and temperature-dependent cases. In the corresponding tornado plot, the colors correspond to the increasing/decreasing parameter value, while the length of the box and its orientation indicates the variation in the number of *at risk* pixels (%) and its direction (right means an increase, left a decrease). The tornado plots provide information on the sensitivity of the model to variations in each individual epidemiological parameter and how the *at risk* area could vary as a consequence. In addition, based on the magnitude of the variation, it helps identify the parameters that most affect our predictions. In the temperature-independent case, the analysis (Figure 6a) shows that the biting rate of *Culicoides* on small ruminants ($a_{s})$ has the most effect on the variation in *at risk* pixels. As a result, the higher the number of bites received by small ruminants, i.e., the higher the number of *Culicoides* who fed on a host, the higher the risk of the spread of Bluetongue. The second most important parameter is the transmission probability from host to vector ($P_{hv}$) whose increase and decrease would result in a 16.6% and −13.6% variation of the pixels *at risk*, respectively. This follows from the fact that increasing the probability means increasing the fraction of infected *Culicoides* in the area. The third most important parameter is the daily mortality rate of vectors ($d_{v}$): A 10% decrease or increase in this mortality rate would result in a variation of −10.3% and 15.2%, respectively. In the temperature-dependent case (Figure 6b), the vector-latency rate ($\alpha_{v}$) is the most important factor. An increase of the rate, and consequently, a reduction of the incubation period, would correspond to a 9.6% increase of the *at risk* pixel, whereas a reduction of the rate would correspond to an 11% reduction of the pixels *at risk*. This follows from the fact that for the same life span, reducing the incubation period implies a longer infection period, and consequently, increases the possibility of infecting the hosts. The biting rate ($a_{s})$ and the vector mortality ($d_{v}$) affect the number of pixels *at risk*. In both cases, the variation in the abundance of the vector population (at fixed host population) has less effect than other epidemiological factors. Consequently, one could think that seasonal variations in vector abundances have less influence less on the extension of the *at risk* area than other parameters.

### 3.3. Estimating the Risk of the Occurrence of BT Outbreaks

After estimating the reproduction number and combining information on livestock mobility and prevalence in Senegal, we estimated the probability of an outbreak of BT being triggered in the area according to Equation (3), (Figure 7). The white area corresponds to areas for which we had no information on mobility. For $R_{0}$ we considered both temperature-independent and temperature-dependent parameters. In both cases, the probability of an epidemic being triggered is high throughout the southern part of Senegal (Casamance and Kédougou region). As mentioned in the previous section, the two transmission maps revealed discrepancies in certain areas, for example, in areas bordering Gambia. However, due to the large flows of incoming animals, the probability of occurrence in these areas, estimated by the two methods, is comparable. We also note that in the Ferlo area, in the northern part of the Senegal River Valley in certain parts of the Groundnut Basin characterized by a low presence of *Culicoides* and low values of $R_{0}$, there are nevertheless areas where the risk of outbreak occurrence is very high (Figure 7). These results can be explained by the large inflows of animals from infected areas. 

## 4. Discussion

The production of transmission maps required modeling the abundance of vectors at the scale of Senegal. To this end, we chose the random forest technique using bioclimatic and environmental variables, since this method is a robust ensemble learning technique for data analysis [41,42,43]. Moreover, in similar work on species abundance in Senegal, Diarra et al. [20] reported that prediction errors between observed and predicted abundances were lower with the RF model than with the generalized linear model (GLM). 

Analysis of abundance maps obtained from random forests showed an increasing abundance of all four species from north to south. This is in accordance with the results of the above-mentioned study by Diarra et al. [20], although they used different explanatory variables. In addition, the predicted abundances of *C. enderleini* are very high in the southern third, and the predicted abundances of *C. imicola* and *C. miombo* are high in both the southern and middle third of Senegal. The predicted abundances of *C. oxystoma*, are high in the southern and western parts of the country. Among the four species studied, *C. imicola* is the only species present along the Senegal River Valley in the northern part of the country. The importance of bioclimatic and environmental variables was underlined by the random forest models and showed that altitude (dem), seasonality of temperatures (Bio04), cattle density (Cattle), and minimum temperature of the coldest month (Bio06) are the variables that most influence the abundance of *C. imicola*, *C. oxystoma*, *C. enderleini*, and *C. miombo*, respectively. However, a previous study by Ciss et al. [21] using Ecological Niche Factor Analysis (ENFA) [44] showed that the presence of *Culicoides* vectors of BTV was negatively correlated with altitude, which, according to the MaxEnt [45] and BRT models, was the most important driver.

Previous studies [19] found high abundances of *C. oxystoma* in areas densely populated by horses, whereas in our study, this variable was not among the 10 variables that most influence the abundance of this species. Our results could be improved if more recent and accurate data on animal distribution were used. In fact, we used data on animal density that dates back to 2006, and these estimates could have changed during the last 15 years.

The data used in this work for the prediction of abundance were obtained from the national trapping campaign carried out between October and September. This means that the predicted abundances do not account for the seasonality of *Culicoides* that could change between trapping periods, as well as from one year to another.

We further analyzed spatial variations in the risk of transmission and occurrence of BT outbreaks in Senegal using species abundance. Previously published maps of the risk of Bluetongue were derived from statistical models of distribution and abundance of vectors [20,25]. The transmission dynamics are explicitly described in our approach, and both the epidemiological parameters and $R_{0}$ have a clear biological interpretation. By combining entomological, climatic, environmental, and small ruminant data, we have produced the first maps of the risk of transmission of the disease in Senegal. Knowledge of the spatial distribution $R_{0}$ can help identify areas where the epidemics could occur ($R_{0}$ > 1) or are unlikely to occur ($R_{0}$ < 1) after the pathogen is introduced into the area from outside.

Moreover, the often-adverse effects of temperature on transmission parameters were included in the estimation of $R_{0}\;$ that could affect the geographical distribution. The results of the temperature-dependent case and temperature-independent case are comparable. Our analysis showed that the risk is low in most areas, but is particularly high in the southern part of Senegal. The risk maps show that the probability of a major epizootic is high in Casamance (Ziguinchor, Sédhiou, and Kolda regions) and Kédougou. In Casamance, the high risk of transmission could be related to the high abundance of *C. enderleini.* Nonetheless, there are several important sources of potential errors in the risk maps presented here. First, vector density estimates are based on measurements made at 96 sites in Senegal (excluding the regions of Ziguinchor and Sédhiou), which introduces a bias in the prediction of vector abundance. Moreover, the model does not account for uncertainty related to the seasonality of the vectors. Indeed, the trapping campaign took place between September and October 2012, because this period corresponded to the peak of *Culicoides* abundance in the Niayes zone, which is not necessarily the case in the other zones of Senegal. The temporal variation in vector abundance affects the risk of transmission and could extend or reduce the *at risk* area by 10%. Many of the epidemiological parameters were taken from the literature, mainly for outbreaks that occurred in European countries, where the contexts differ from that in Senegal. We used sensitivity analysis to explore possible scenarios, due to variations in parameters. Overall, the analysis showed that under- or over-estimation of the daily bite rate on small ruminants, the vector incubation period, and the transmission probability from host to vectors, have a strong influence on the basic reproduction number $R_{0}$, with the first two parameters depending on environmental conditions. We noted that a 10% increase/decrease in the biting rate would result in an increase in the basic reproduction number $R_{0}\;$ of 9.56% and −8.01%, respectively. To improve our estimates, the model needs to be calibrated with recent field data (serological or outbreak reports) to provide more reliable estimates of the biting rate and transmission probabilities.

Knowledge of $R_{0}$ makes it possible to identify areas where the disease can occur, where an infected animal or vector is introduced, and in some cases, could provide information on the attack rate. However, the probability that no outbreak occurs after the introduction of a single infected animal is proportional to the reciprocal $R_{0}$ (i.e., 1/$R_{0}$). We estimated the probability of occurrence by combining mobility and prevalence data from the 2018 national sero-survey. The results confirmed our previous conclusions drawn from transmission maps: Areas with high values of R_0_ were likely to experience outbreaks with a high probability of occurrence in the south. However, the analysis of the map shows that livestock mobility could play an important role in triggering epidemics. In areas with low $R_{0}$ values (between 1 and 2), an epidemic can still occur if enough infected animals are introduced. This is the case in the Ferlo area and at the border with Gambia where there is a high probability of occurrence, despite their low $R_{0}$ values. Therefore, despite the fact that BT is a non-contagious disease, livestock mobility can play an important role in its diffusion. To validate our results, we need more information on the epidemiological situation in the region: In fact, we are unable to distinguish if our seroprevalence results are related to epidemic dynamics or low noise circulation or if the animals that were imported were already infected or not. Although our livestock data include information on international movements, in this article, we focused on national movements, since no information was available on the BT epidemiological situation in neighboring countries. Being able to include such, the information would make it possible to assess the risk of introducing the disease from other countries 

Despite this limitation, the risk maps could help design an effective surveillance system for the control of Bluetongue, for example, by prioritizing vaccination campaigns in high-risk areas.

## 5. Conclusions

Bluetongue is a veterinary disease with a huge impact on animal production. Following outbreaks of AHS in Senegal, which is transmitted by the same vectors, interest in Bluetongue disease has increased. Because most of the symptoms resemble those of other diseases, some cases may be misdiagnosed or not noticed, and the epidemiology of the disease remains unknown today. 

Mobility appears to play an important role in outbreak and disease spread, with large flows of infected animals also triggering epidemics in areas with low values of $R_{0}$. However, despite the high probability of outbreaks occurring in most areas, few (if any) outbreaks are declared to central authorities. This is probably because the disease can appear in a mild form, and is not consequently recognized by farmers, for example. The fact BT is not on the list of the 13 priority diseases in Senegal (These are: Anthrax, Rift Valley Fever, Rabies, New Castle Disease, Highly Pathogenic Avian Influenza, Lumpy Skin Disease, Foot and Mouth Disease, Contagious Bovine Pleuropneumonia, Pasteureullosis, Rinderpest, Peste des Petits Ruminants, African Horse Sickness, African Swine Fever (http://www.fao.org/3/CA2917FR/ca2917fr.pdf)) today could also explain under-reporting. Raising the awareness of livestock owners to the symptoms and effects of the disease could help improve surveillance and control of this disease.

Our work is one of the first steps in assessing the impact of BT in Senegal, but has some limitations. There is now a need for follow-up studies of the abundance of vectors to provide information concerning variations over the year and assess vector’s competence.

## Figures and Tables

**Figure 1 microorganisms-08-01766-f001:**
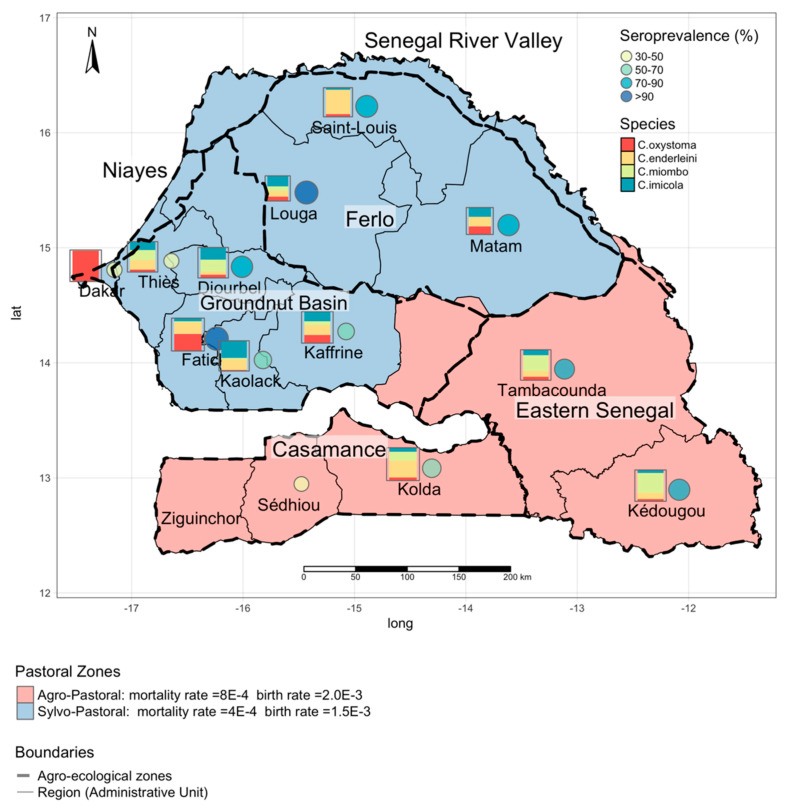
Seroprevalence, entomological, and demographic data. The circles show regional BT seroprevalence based on data collected during the national sero-survey in 2018. The squares show the distribution of the four *Culicoides* species of interest; the square’s size is proportional to the logarithm of the number of *Culicoides* collected during the national entomological survey in 2012. The background colors (red and blue) delimit the two pastoral zones with different birth and mortality rates. Dashed lines delimit the six agro-ecological zones in Senegal (Ferlo, Senegal River Valley, Eastern Senegal, Casamance, Groundnut Basin, Niayes), while the thin solids lines identify the administrative regions.

**Figure 2 microorganisms-08-01766-f002:**
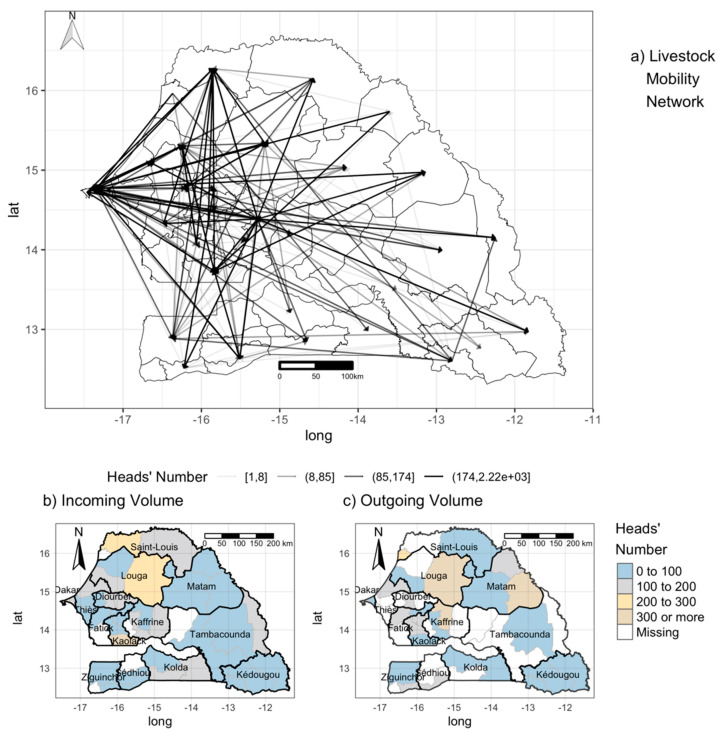
Mobility data. On top (**a**) representation of livestock mobility network, each line corresponds to movements existing between the department, and the color is relative to the number of animals exchanged. The bottom part, departments are colored based on: (**b**) The incoming volume (number of animals arriving in the department); (**c**) outgoing volume (number of animals leaving the department), na: data not available.

**Figure 3 microorganisms-08-01766-f003:**
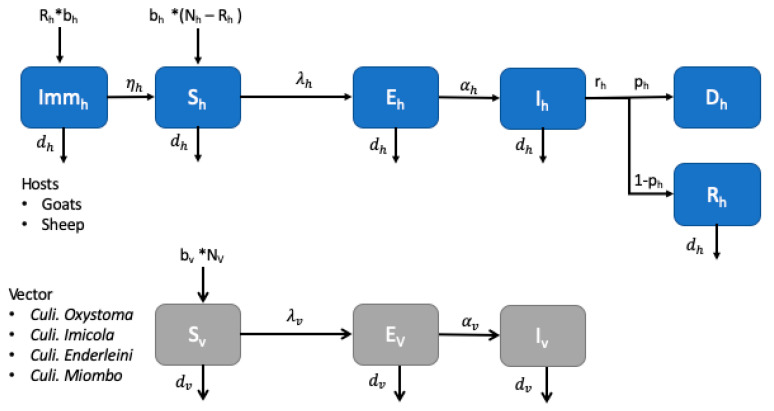
Pictorial representation of the mathematical model. Host dynamics are at the top of the figure, vector dynamics at the bottom.

**Figure 4 microorganisms-08-01766-f004:**
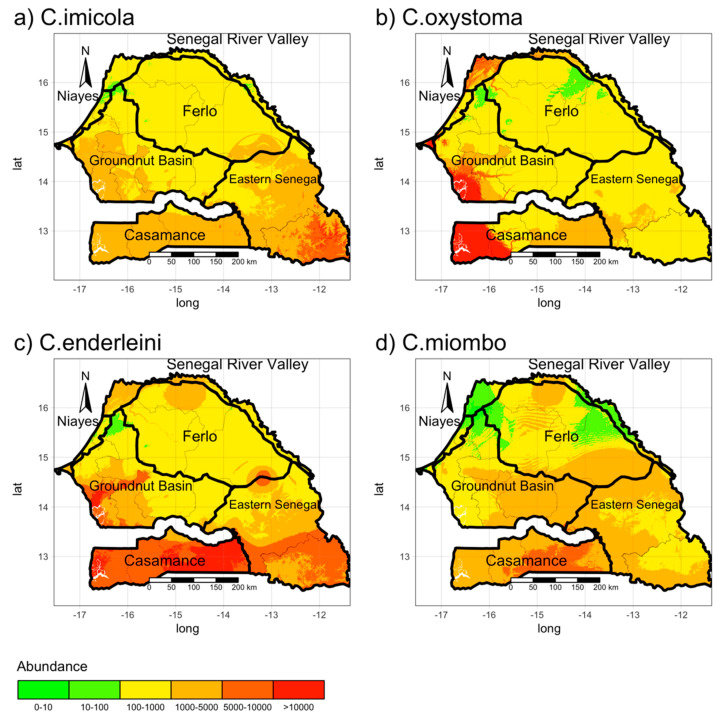
Maps of abundance of the four main Bluetongue virus (BTV) vector species in Senegal: (**a**) *C. imicola*; (**b**) *C. oxystoma*; (**c**) *C. enderleini*, and (**d**) *C. miombo*.

**Figure 5 microorganisms-08-01766-f005:**
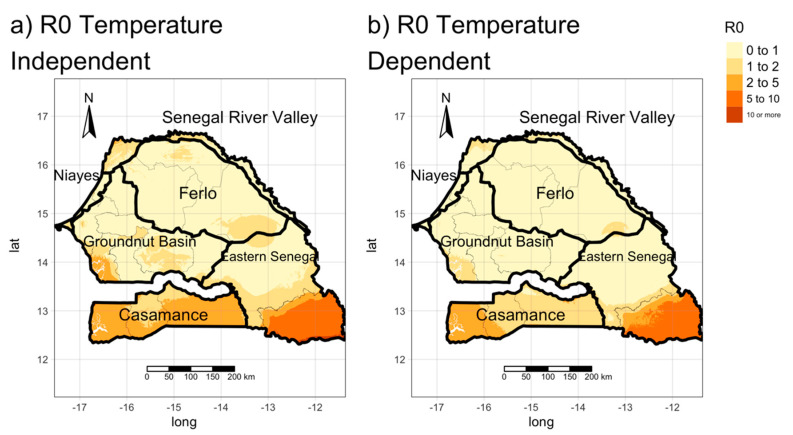
Transmission maps ($R_{0}$) (**a**) realized using temperature-independent parameters; (**b**) using temperature-dependent parameters.

**Figure 6 microorganisms-08-01766-f006:**
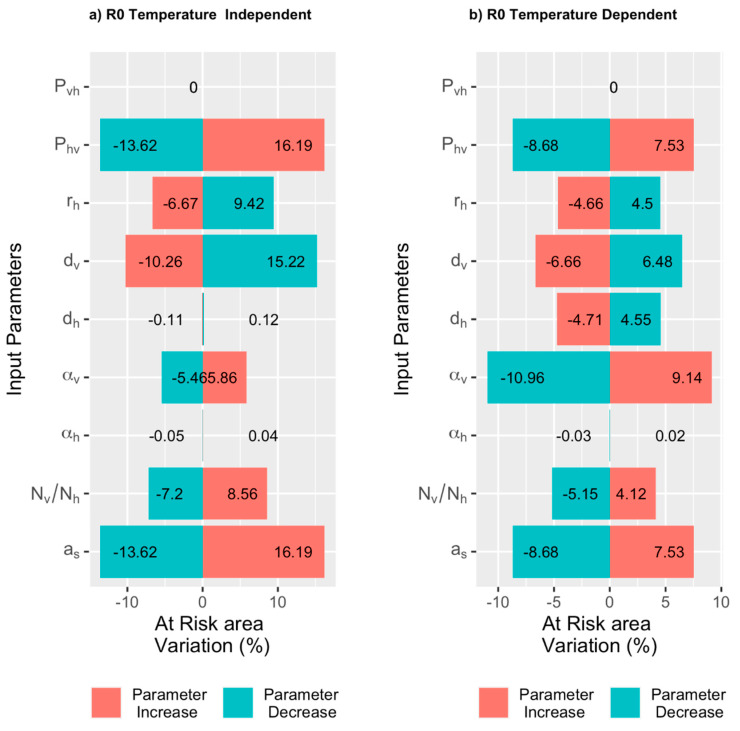
Tornado plot of sensitivity analysis in the 2 cases when parameters used in the model are temperature-independent (**a**) or temperature-dependent (**b**). The name of the epidemiological parameters that are varied on shown on the y-axis. The variation in the number of pixels *at risk* is shown on the x-axis. For each parameter, the bars represent the variation in the number of *at risk* pixels following a decrease or increase in a parameter value. The color of the bar indicates if the variation in the number of pixels *at risk* is due to an increase (red) or a decrease (blue) in the corresponding parameter value. The length of the bar is proportional to the variation in the number of pixels *at risk* (expressed as a percentage).

**Figure 7 microorganisms-08-01766-f007:**
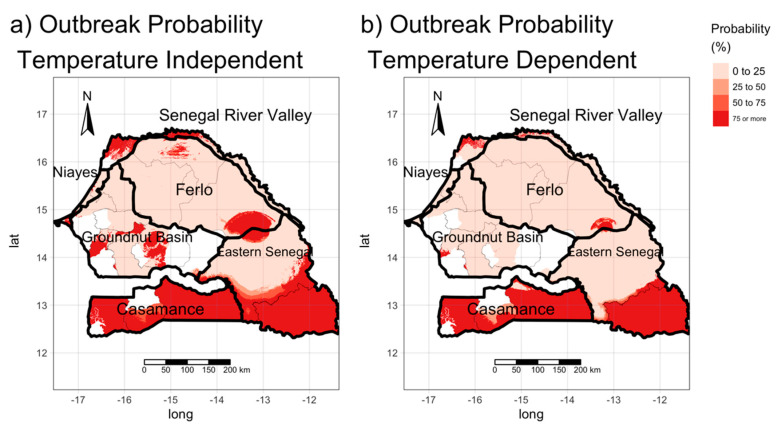
Probability of outbreaks due to the introduction of infected animals. (**a**) using R_0_ temperature-independent values, (**b**) using *R*_0_ temperature-dependent values. Livestock movements from infected areas could explain why areas with a low value of *R_0_* could have a high risk of outbreak occurrence. White areas correspond to areas where the outbreak probability has not been estimated because of missing information.

**Table 1 microorganisms-08-01766-t001:** Bluetongue epidemiological parameters.

Parameter	Description	Temperature Independent	Temperature-Dependent
		Values	References	Values	References
$d_{v}$	Vector mortality rate	0.16 [0.1–0.5]	[36,37]	$0.009e^{0.16T}$	[36]
b_v_	Vector fertility rate	6.1 eggs/year	[38]		
α_v_	Latency rate for vector	0.09 [0.06–0.1]	[33,36]	0.0003T(T−10.4)	[35,36]
P_vh_	Transmission Vector-host	0.9 [0.8–1.0]	[36]		
a_s_	Biting rate	0.17 [0.05–0.4]	[36]	0.0002×T×(T−3.7)×(41.9−T)127	[36]
d_h_	Host mortality rate	0.0005 (Sylvo-pastoral area)0.0008 (Agro-pastoral area)	[39]		
b_h_	Host fertility rate	0.002 (Sylvo-pastoral area)0.0015 (Agro-pastoral area)	[39]		
α_h_	Latency rate for host	0.0625	[33]		
r_h_	Recovery rate for host	0.125	[27,36]		
d_h_	Host mortality rate	0.0005 (Sylvo-pastoral area)0.0008 (Agro-pastoral area)	[39]		
b_h_	Host fertility rate	0.002 (Sylvo-pastoral area)0.0015 (Agro-pastoral area)	[39]		
α_h_	Latency rate for host	0.0625	[33]		
r_h_	Recovery rate for host	0.125	[27,36]		
P_hv_	Transmission Host Vector	0.05 [0.001–0.15]	[36,37]		
p_h_	Disease induced mortality	0.01 [0.001–0.01]	[36]		
ηh	Rate of immunity loss	1/90	[39]

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
