# Peer review of "Assessing the Risk of Occurrence of Bluetongue in Senegal"

_microorganisms, 2020, doi:10.3390/microorganisms8111766_

Round 1

Reviewer 1 Report

In abstract, the latter parts of the manuscript have not been fully integrated. I suggest rewrite lines 34-36 as follows: R0 <1 for most areas of Senegal. The southern (Casamance) and south-eastern (Kedougou and part of Tambacounda) agro-pastoral areas have the highest risk of outbreak, which is temperature independent.  The next higher risk of outbreaks are in areas of the Senegal River Valley and the Atlantic coasts. Seroprevalence rates, shown by cELISA, were not positively correlated with the outbreak probability (better supported with a statistics - correlation of seroprevalence rate with Fig. 7).

Fig. 1: Figure 1 is much improved, but can be further improved. Enlarge Fig. 1, and incorporate the mobility data (by different sizes of the arrow and by directions). This will make Fig. 1 more "dynamic", because these high risk areas bordered with several neighboring countries like Mali, Gambia, Guine-Bissau, and Guinee.  The colors you use to represent Culicoides species are "not distinctive".  I suggest use white background, yellow color for C. enderleini and dark blue color for C. imicola, or whatever you think is most distinctive to separate these four.  Proportionally enlarge the marks of seroprevalence and Culicoides in the map.

Fig. 2:   This presentation looks rather "static". It is difficult to figure out the outgoing and incoming volume by comparing these two figures.  I suggest using outgoing by one color, and incoming by another color. Mark each area with different color intensities to represent different volume.  

Figs. 4 and 7: I think it is possible to analyze the correlation of Culicoides species intensities with the outbreak probability.  If you can do the statistics, the results will be even more significant.

Author Response

Dear Reviewer, we'd like to thank you for all your comments and suggestions. We tried our best to  fulfill your requests

Please see attachment for our answers 

Reviewer 2 Report

The Manuscript of Gahn described a risk assessment of BTV outbreak potential in Senegal. The paper reads well and has a relatively simple conclusion, despite the thorough analysis performed within.

I only have a few minor points:

Line 28: clarify that BTV is implied by "its distribution" and not Culicoides.

Lines 35-36: it would be good to state R in each of these areas, as the abstract seems a bit light on actual results.

Line 59: the reference of Roy  here is not entirely suitable. At least say that the outbreaks in these areas are reviewed in this paper, if so. Otherwise, please provide a better reference.

Line 63: This should be rephrased. It also may be worth including the number of serotypes detected to date.

Line 81: "pathogenic agents"- please just state BTV and AHSV

Figure 1: Should the axes not be defined?

Figure 1: the !"boxes" should be made larger, or presented in a different way. For instance, thee tiny box near Louga is unhelpful. 

Figure 1. The placement of the figure is confusing as it is not clear where this data has come from.

Line 114: I counted 16 regions on the figure, whoever, I may be mistaken!

Line 131: why is there a capital V and I?

Line 131: This sentence needs a full stop.

Line 179: Reference?

Line 195: Spelling error "provideing"

Figure 2: where is na used in the figure?

Line 306: improve English for "each pixel of Senegal map..."

Line 317. add "of" in "little risk an outbreak"

Line 330: Global sensitivity? 

Section 3.3: mentioning figure 5 after figure 7 is a bit jarring. 

Line 477: What are the 13 disease or please provide a reference. 

Author Response

Dear Reviewer,

thank you very much for your feedback on our work

Please see the attachment for our answers 

Bests

Andrea
